# Contribution of Coagulase and Its Regulator SaeRS to Lethality of CA-MRSA 923 Bacteremia

**DOI:** 10.3390/pathogens10111396

**Published:** 2021-10-28

**Authors:** Ying Liu, Wei Gao, Junshu Yang, Haiyong Guo, Jiang Zhang, Yinduo Ji

**Affiliations:** 1Department of Veterinary and Biomedical Sciences, College of Veterinary Medicine, University of Minnesota, Saint Paul, MN 55108, USA; liuyd@shafc.edu.cn (Y.L.); yang1181@umn.edu (J.Y.); guohaiyong78@163.com (H.G.); 2Department of Biomedicine and Health Sciences, Shanghai Vocational College of Agriculture and Forestry, Shanghai 201699, China; 3Key Laboratory of Jiangsu Preventive Veterinary Medicine, College of Veterinary Medicine, Yangzhou University, Yangzhou 225009, China; wgao@yzu.edu.cn; 4School of Life Science, Jilin Normal University, Siping 136000, China

**Keywords:** *Staphylococcus aureus*, CA-MRSA, coagulase, SaeRS regulator, virulence, bacteremia

## Abstract

Coagulase is a critical factor for distinguishing *Staphylococcus aureus* and coagulase-negative Staphylococcus. Our previous studies demonstrated that the null mutation of coagulase (*coa*) or its direct regulator, SaeRS, significantly enhanced the ability of *S. aureus* (CA-MRSA 923) to survive in human blood in vitro. This led us to further investigate the role of coagulase and its direct regulator, SaeRS, in the pathogenicity of CA-MRSA 923 in bacteremia during infection. In this study, we found that the null mutation of *coa* significantly decreased the mortality of CA-MRSA 923; moreover, the single null mutation of *saeRS* and the double deletion of *coa*/*saeRS* abolished the virulence of CA-MRSA 923. Moreover, the mice infected with either the *saeRS* knockout or the *coa*/*saeRS* double knockout mutant exhibited fewer histological lesions and less neutrophils infiltration in the infected kidneys compared to those infected with the *coa* knockout mutant or their parental control. Furthermore, we examined the impact of *coa* and *saeRS* on bacterial survival in vitro. The null mutation of *coa* had no impact on bacterial survival in mice blood, whereas the deletion mutation of *saeRS* or *coa*/*saeRS* significantly enhanced bacterial survival in mice blood. These data indicate that SaeRS plays a key role in the lethality of CA-MRSA 923 bacteremia, and that coagulase is one of the important virulence factors that is regulated by SaeRS and contributes to the pathogenicity of CA-MRSA 923.

## 1. Introduction

*Staphylococcus aureus* is a critical human pathogen that can often cause skin and soft tissue infections and sometimes leads to serious systematic infections such as pneumonia, endocarditis, and toxic shock syndrome [1,2]. The continuous emergence of methicillin-resistant *S. aureus* (MRSA), including hospital-acquired (HA)-MRSA and community-acquired (CA)-MRSA has caused public health concerns worldwide [3]. Most MRSA isolates are resistant to multiple low-cost antibiotics, which in turn limits therapeutic choices to treat MRSA-associated infections [4]. Therefore, a better understanding of the pathogenesis of *S. aureus*, including HA-MRSA and CA-MRSA, enables us to develop an alternative strategy to combat MRSA infections. 

The ability of *S. aureus* to cause infections is partially dependent on the expression of virulence factors such as protein A, fibronectin-binding proteins, coagulase, cytotoxins, proteases, DNases, and superantigens [5], which allow the bacteria to invade host cells and evade host immune systems [6,7]. The expression of these virulence factors is coordinately regulated by an array of regulators, including two-component regulatory systems (TCSs) such as Agr [8,9,10], SaeRS [11,12,13,14,15,16], and ArlRS [17,18,19], as well as global transcriptional regulators such as SarA [20,21,22,23], SarZ, and MgrA [24,25].

It is well established that SaeRS regulates the expression of critical virulence genes, such as *hla*, *hlb*, *hlgABC*, *lukED*, *fnbA*, and *coa* [11,12,14,26,27,28,29]. The disruption of the SaeRS signaling pathway remarkably impaired the cytotoxicity of *S. aureus* and its capacity to internalize into a variety of host cells [12,28]. The null mutation of *saeRS* attenuated the pathogenicity of *S. aureus*, including HA-MRSA WCUH29 and USA300 CA-MRSA 923 strains in the animal models of infection [9,12,13,26,30,31].

SaeRS controlled the transcription and expression of staphylococcal coagulase through the direct binding of the response regulator SaeR to the promoter regions of *coa* [16]. Coagulase is able to convert host prothrombin to staphylothrombin, which in turn activates the protease activity of thrombin. It was postulated that the coagulase could cause localized clotting and consequently enable bacterial pathogens to escape from phagocytic and immune defenses. However, reports on the role of coagulase in the pathogenicity of *S. aureus* in animal models of infection have been contradictory [32,33,34]. 

Our previous studies demonstrated that the null mutation of coagulase (*coa*) or its direct regulator, SaeRS, significantly enhanced the ability of *S. aureus* (CA-MRSA 923) to survive in human blood in vitro [35]. This led us to further elucidate the role of coagulase in the pathogenicity of CA-MRSA 923 in in vivo blood infection. In this study, we determined the importance of coagulase and its direct regulator, SaeRS, for the pathogenicity of CA-MRSA 923, using a mouse model of blood infection as well as using histopathology. Meanwhile, we examined the role of coagulase and SaeRS in the survival of bacteria in mouse blood. The findings enabled us to pinpoint the role of coagulase and SaeRS in the pathogenesis of CA-MRSA 923 strain-caused bacteremia in a mouse model of blood infection.

## 2. Results

### 2.1. The Deletion Mutation of coa Significantly Alleviated the Lethality of CA-MRSA 923 in a Murine Blood Infection

It has been reported that coagulase is an important virulence factor of *S. aureus* [34]. However, our previous study demonstrated that coagulase impaired the survival of CA-MRSA 923 in human blood [35]. This controversy between in vivo and in vitro blood infection led us to further determine the role of coagulase in the pathogenicity of CA-MRSA 923 using a mouse model of infection. We performed a double-blind animal study. The mice were infected with the bacterial cells of exponential phase culture via tail vein injection (see Table 1); the mortality of infected mice was calculated every 24 h after infection. All surviving mice infected with the bacterial cells exhibited clinical features of infection, including ruffled fur and grayish color as compared to non-infected control mice (data not shown). 

The wild-type control, CA-MRSA 923, caused 91% death of mice (10/11 mice in Trial I (n = 6) and Trial II (n = 5)) on day 3 after infection; in contrast, no mice with the *coa* deletion mutant (Trial I (n = 6) and Trial II (n = 5)) died from infection (Figure 1). On day 4, 100% of the wild-type control mice (n = 5) died from infection, whereas 80% and 60% of the mice with the *coa* deletion mutant (n = 5) survived after day 5 and 7 of infection, respectively. Moreover, there was a 20% difference of mortality rates observed between the mice infected with the *coa* and *saeRS* mutant or *saeRS*/*coa* double mutant on day 7. In addition, the bacterial burden significantly decreased in the kidneys and spleens of mice 3 days after infection with *coa* mutant compared to those infected with the wild-type control (Figure 2A,B). Taken together, these results clearly indicated that coagulase is an important virulence factor for the CA-MRSA 923 strain to cause lethal infection of Balb/c mice.

### 2.2. The saeRS Single Knockout and coa/saeRS Double Knockout Significantly Eliminated the Lethality of CA-MRSA 923 in a Murine Blood Infection

It has been demonstrated that the SaeRS two-component system plays a critical role in the pathogenesis of *S. aureus* in various models of infection [9,12,13,30]. Of particular importance, it was reported that SaeRS contributed to the pathogenesis of the CA-MRSA 923 strain in a mouse lung infection [9]. However, our previous studies also indicated that SaeRS is an anti-phagocytic regulator through the mediation of coagulase production, since the deletion mutation of saeRS significantly elevated the anti-phagocytic capacity of CA-MRSA 923 in human blood in vitro, whereas overexpression of coa eliminated the enhanced survival phenotype of the SaeRS null mutant [35]. These results led us to further investigate the role of SaeRS in the pathogenesis of CA-MRSA 923.

The mice infected with either a *saeRS* null mutant, *coa*/*saeRS* double knockout mutant, or their parental control exhibited clinical features of infection, including ruffled fur and grayish color compared to the non-infected control (data not shown). Moreover, 100% and 80% of the mice (n = 5) survived 5 and 7 days, respectively, after infection with either SaeRS or Coa/SaeRS null mutant, whereas none of the mice (n = 5) survived 5 days after infection with wild type CA-MRSA 923 (Figure 1). 

Furthermore, the bacterial burden significantly decreased in the kidneys and spleens of mice 3 days after infection with either *saeRS* knockout mutant or *coa*/saeRS double knockout mutant, compared to those infected with the wild type control (Figure 2A,B). In addition, the *coa/seaRS* double mutation remarkably alleviated bacterial burden in the kidneys of mice, compared to those infected with the *coa* single deletion mutant (Figure 2A). These results clearly indicate that SaeRS is a critical virulence regulator, and its regulated coagulase is a major contributor to the lethality of CA-MRSA 923-caused bacteremia in the mouse model of blood infection.

### 2.3. The Deletion Mutation of saeRS or coa/saeRS Remarkably Diminished the Histologically Pathological Damages in the Kidneys of Mice Infected with CA-MRSA 923

In order to further elucidate the roles of coagulase and its direct regulator, SaeRS, we conducted histopathological image analysis of infected kidney tissues. The partial kidneys isolated from infected mice were fixed and sliced. Hematoxylin and eosin staining showed that more neutrophils infiltrated into the kidneys of mice that are infected with wild-type strain compared to those non-infected control, and those infected with *coa*/*saeRS*, *saeRS*, or *coa* knockout mutant (Figure 3). No histological lesions were observed in the kidneys in the negative control (E2). However, varying degrees of renal tubular epithelial cell swelling and even necrosis were observed in other groups. Particularly, the group A mice that were infected with the wild-type CA-MRSA 923 strain exhibited the most serious symptoms of infection (Figure 3A): (i) the scattered bacterial communities (long arrow) were observed in the kidneys 3dpi (Figure 3A1); (ii) swelling or necrosis of renal tubular epithelial cells occurred 4dpi (Figure 3A2); (iii) simultaneously, severe abscesses could be observed in the infected kidneys; (iv) moreover, numerous bacterial communities were surrounded by a certain amount of neutrophil infiltration (short arrow) and hemorrhage, separated from neutrophils by an amorphous pseudo capsule (Figure 3A1,A2). 

In the *coa*-knockout mutant infected mice (Figure 3B), (i) a few infiltrations of neutrophilic granulocytes were observed 3dpi (Figure 3B1); (ii) the distribution of bacterial communities were observed 7dpi (Figure 3B2), and (iii) in the abscesses, more neutrophils were observed than those in A group; however, the slight pathological changes of mice infected with either *coa*/*saeRS* knockout mutant or *saeRS* knockout mutant were observed under a microscope, and only a small number of renal tubules were filled with neutrophils (Figure 3C1–D2). 

### 2.4. The Deletion Mutation of saeRS Significantly Increased, but Coagulase Null Mutation Had No Influence on the Ability of CA-MRSA 923 to Survive in Mouse Blood In Vitro

To elucidate whether the role of *saeRS* and *coa* in the pathogenesis of *S. aureus* in the mouse model of blood infection is the same when in vitro in mouse blood, we performed phagocytosis assays using the pooled blood from three mice. The results showed that the *saeRS* and *saeRS*/*coa* double null mutation remarkably enhanced the survival capacity of the *S. aureus* CA-MRSA 923 strain. There were much more bacteria that survived in the mouse blood after 1 h of infection with the *saeRS* and *saeRS*/*coa* double null mutations compared to the wild-type control or the *coa* knockout mutant (Figure 4). However, the *coa* null mutation had no remarkable effect on bacterial survival in the mouse blood (Figure 4). 

To further characterize the *coa*, *saeRS*, and *coa*/*saeRS* double mutants, we performed a blood coagulation assay and observed that the deletion of *coa*, *saeRS*, and *coa*/*saeRS* obviously eliminated blood coagulation compared to the wild type control CA-MRSA 923 (Figure 5). We also performed complementation studies and observed that the introduction of plasmid pYH4-*coa* into the *coa* or *coa*/*saeRS* double mutant restored the blood coagulation capacity (Figure 5). 

## 3. Discussion

Our results clearly indicated that coagulase (*coa*) contributes to the virulence of CA-MRSA 923 bacteremia in a mouse model of infection. The deletion mutation of *coa* significantly decreased the mortality of infected mice; however, the *coa* knockout strain still induced obvious pathological changes, including abscess formation and infiltration of neutrophils to the kidney tissue, when compared to the negative control, as well as *saeRS* single or *coa/saeRS* double knockout infected mice (Figure 3). These data suggest that coagulase might be one of the important virulence factors of the USA300 CA-MRSA 923 strain in a mouse model of bloodstream infection; moreover, coagulase might function differently in *S. aureus* bacteremia and local organ infection. 

In addition, in vitro studies showed that coagulase (*coa*) had no obvious effect on the survival of CA-MRSA 923 in the mouse blood, which is inconsistent with our previous in vitro studies that demonstrated that coagulase (*coa*) could increase the bacterial susceptibility to phagocytic killing in human blood [35]. This disparity suggests that coagulase might employ different mechanisms to enable *S. aureus* survival in a systematic infection and in vitro blood infection, as well as in the different species of blood infection. However, it is unclear whether the addition of heparin in blood affects the function of staphylococcal coagulase during in vitro incubation in blood. Moreover, we do not know whether the expression level of coagulase is different between in vivo bloodstream and in vitro blood infection. To better elucidate the pathogenesis of the USA300 CA-MRSA 923, it is necessary to further explore the molecular and cellular basis of coagulase function during infection in future studies.

There are contradictory reports about the role of coagulase as a virulence factor. In one case, coagulase did not affect the pathogenicity of *S. aureus* in a rat endocarditis model of infection [32,33]; in addition, coagulase-negative staphylococci (CNS) induced renal abscesses and lethality in a mouse model of bloodstream infection [36,37]. On the other hand, our study is partially consistent with other group’s studies that demonstrated that coagulase played a crucial role in the pathogenesis of *S. aureus* in a mouse model of bloodstream infection, despite the fact that different *S. aureus* strains were used in these studies [34]. However, in our studies, 60% of the mice could survive 7 days after infection with the *coa* knockout mutant of CA-MRSA 923 (Figure 1), whereas no mice survived 3 days after infection with the *coa* knockout mutant of Newman strain [34]. This disparity is probably due to the different genetic backgrounds of the CA-MRSA 923 and Newman strains. It was revealed that there is a point mutation in the histidine kinase SaeS, the sensor of the *coa* regulator SaeRS in the Newman strain, which led to increased Eap-dependent cellular invasiveness [38]. Moreover, our previous studies revealed that the deletion of *coa* totally eliminated the coagulation activity of CA-MRSA 923 strain. In contrast, the deletion mutation of *coa* could not abolish the coagulation activity of the WCUH29 strain [35,39], suggesting the expression of other coagulation factors, such as the von Willebrand factor binding protein (vWbp) in WCUH29 strain [40]. Indeed, it was revealed that both coagulase (*coa*) and vWbp contribute to the clotting of mouse blood for the Newman strain, and vWbp is also a critical virulence factor for Newman strain induced bacteremia [34]. Therefore, without comprehensive determination of the expression levels of coagulation factors, including coagulase and vWbp in various USA300 CA-MRSA isolates, we cannot conclude that the importance of coagulase for the lethality of CA-MRSA 923 strain-caused bacteremia is universally applicable to other CA-MRSA strains. 

Our study also demonstrated that the two-component regulator SaeRS, which is a direct regulator of *coa* transcription, is a key virulence regulator for CA-MRSA 923-caused bacteremia in the mouse model of blood infection. This result is consistent with previous reports using different animal models of infection [9,12,13,30]. It has also been shown in the same strain (CA-MRSA 923) that SaeRS affects virulence in the mouse models of necrotizing pneumonia and skin infections [9]. Consistent with the mortality results, our histopathological analyses also clearly indicated that the two-component regulator, SaeRS, is a key virulence regulator of *S. aureus*, because *saeRS* single or *coa*/*saeRS* double knockout remarkably affected the pathogenicity of CA-MRSA 923. The kidney tissues of mice infected with either *saeRS* or *coa*/*saeRS* knockout mutant exhibited minor pathological changes compared with those infected with either the parental control or the *coa* knockout mutant. Moreover, the *coa*/*saeRS* double mutant significantly decreased the bacterial burden in the kidneys of infected mice compared with those infected with the *coa* knockout mutant. This is likely attributable to the multiple virulence factors that are regulated by SaeRS, including coagulase in CA-MRSA 923 [7] and other *S. aureus* strains [16,29]. It is well established that *S. aureus* infection includes several key steps, such as colonization, proliferation, evasion of innate defenses, and cytotoxicity [41,42,43,44], and that multiple factors contribute to a key step of bacterial survival in blood [45]. SaeRS controlled the expression of adhesins and invasin molecules such as fibronectin-binding proteins [46], which in turn regulate the internalization of human epithelial and endothelial cells by *S. aureus* [11,12,47], as well as mediate the expression of toxins, important cytotoxic and lethal factors of *S. aureus* [12,16,44,48,49]. It is unclear whether SaeRS mediates the expression of coagulase in mouse blood during infection.

Conversely to mouse bloodstream infection studies, *saeRS* null mutation remarkably increased the bacterial survival in the mouse blood in vitro, which is consistent with our previous findings in human blood [35]. Our previous studies demonstrated that the expression of Coa and SaeRS *in trans* restored the bacterial survival of *coa* and *saeRS* deletion mutant, respectively, in human blood [35]. Although the heparin might affect the function of coagulase during in vitro phagocytosis assays, the disparity of the role of SaeRS between in vitro and in vivo blood infection further suggests that CA-MRSA 923 strain might employ different mechanisms between in vitro anti-phagocytosis in blood and in vivo anti-immune systems, which should be further explored in future studies. 

We acknowledge that our study lacked quantitative analysis of the number and size of lesions in the kidneys of infected mice and lacked complementation studies for in animal studies. Feeding animals with a broad-spectrum antibiotic erythromycin to maintain the plasmid (pYH4, pYH4-*coa*, and pYH4-*saeRS*) for 7 days possibly complicated the results of the bloodstream infection. Our studies demonstrated that the complementation of Coa or SaeRS could restore the susceptibility of *coa* or *saeRS* knockout mutant to human blood in vitro [35], and that the introduction of pYH4-*coa* could restore the blood coagulation of the *coa*, *saeRS*, and *coa*/*saeRS* double mutants. However, it is still necessary to determine whether any off-target mutations occurred during the process of generating *coa*, *saeRS*, and *coa/saeRS* deletion mutants, using whole genome DNA sequencing and RNA-Seq analyses in future studies.

In conclusion, we demonstrated that SaeRS plays a key role in the lethality of CA-MRSA 923 bacteremia, and that coagulase is one of the important virulence factors that is regulated by SaeRS and contributes to the pathogenicity of CA-MRSA 923. 

## 4. Materials and Methods

### 4.1. Bacterial Strains and Growth Media 

The bacterial strains used in this study are listed in Table 2. *S. aureus* was incubated in Trypticase Soy Broth (TSB; Difco) with appropriate antibiotics, as indicated for 18 h or 2 to 3 h until the bacteria reached the mid-log phase (~0.4 at OD600 nm) at 37 °C with shaking at 220 RPM or on TSA agar at 37 °C. 

### 4.2. S. aureus Survival and Blood Coagulation Assays 

The bacterial survival assays in blood were conducted as described [6,50,51]. Briefly, the bacterial cells were harvested from the mid-exponential phase of culture in TSB, washed three times with PBS, and suspended in PBS. Fresh venous blood was collected from 3 female Balb/c mice and transferred into heparin-containing tubes. Approximately 1 × 10^7^ CFUs of bacteria were inoculated in 1 mL of heparinized mouse blood and incubated for 1 h at 37 °C with shaking. The initial inoculated bacterial samples, as well as the bacteria and blood mixed samples, were collected, respectively, after 1 h of incubation diluted and plated onto TSA for viable CFUs. Each strain included 3 repeats for the phagocytosis assays. The percentage of surviving bacteria was calculated as (CFU time point/CFU initial input) × 100. 

To assess bacterial blood coagulating activity, Lepirudin-treated human blood was infected with approximately 2 × 10^5^ CFU/mL of CA-MRSA 923 wild type, mutants, or complementary strains and incubated for 24 h at 37 °C. Blood coagulation was assessed by tipping the tubes to 45 ° angles, as described [34].

### 4.3. Mouse Intravenous Infection Model and Histopathology

Five-week-old Balb/c female mice (Shanghai Jihui Laboratory Animal Breeding Co. Ltd, Shanghai, China) were used for the animal studies. The mice (55 total mice) were randomly divided into 5 treatment groups, weighed before the infection, and were used for double blind animal studies. Each treatment was then repeated (Trial I (n = 6 in each treatment group); Trial II (n = 5 in each treatment group)) (Table 2). The cohort of mice was infected with approximately 5 × 10^7^ CFUs of the mid-log phase of bacterial cells via a tail vein injection with a 30-gauge insulin syringe (see Table 2). The negative infection control group of mice was injected with PBS (Table 2). The mice were monitored daily, and the dead mice were collected. The kidneys and spleens of the mice were then isolated for the determination of bacterial burden and histopathological studies. For Trial I infected mice (n = 6 per treatment) and Trial II infected mice (n = 5 per treatment), the surviving mice were euthanized on day 3 and day 7 after infection, respectively, with an overdose of cylinder CO_2_. The mortality of the infected mice on day 3 was calculated by using the pool data from two trials (n = 11). The kidneys and spleens were isolated from the dead mice and sacrificed mice; for histopathology, the part of the isolated kidneys from each treatment group (n = 2) was immersed in 1 mL of 10% neutral buffered formalin, infiltrated with serial increasing concentrations of ethanol and xylene, and then embedded in paraffin. The kidneys were thin sectioned at 10 μm and stained with hematoxylin and eosin using a standard procedure for histopathology examinations under light microscope. To determine the bacterial burden in the infected spleens and kidneys 3 days after infection, the isolated spleens and the kidneys from each treatment group (n = 6) in Trial I were weighed, homogenized, and diluted in PBS containing 0.25% Triton-X100, and plated onto TSA for viable CFU. 

### 4.4. Data Analysis

Independent samples were statistically analyzed using Student’s *t*-test or a Chi-Square with an alpha level ≤ 0.05 considered significant. 

## Figures and Tables

**Figure 1 pathogens-10-01396-f001:**
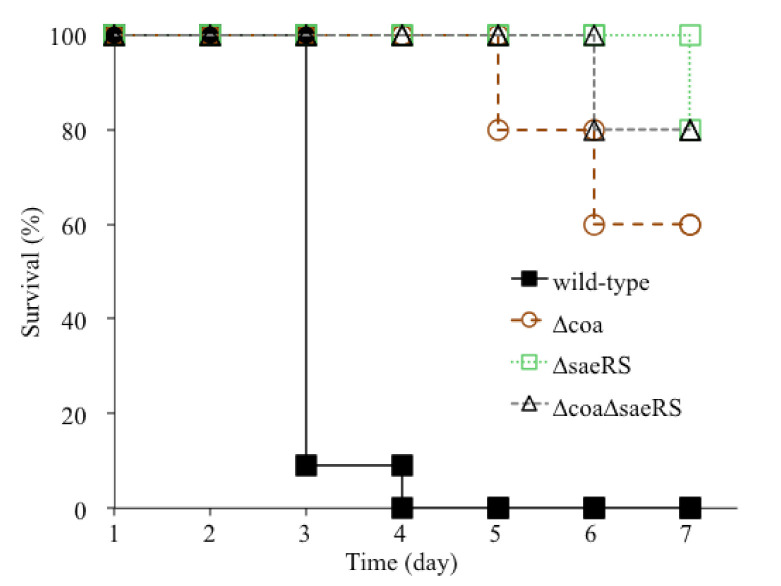
Effect of *coa*, *saeRS*, or *coa*/*saeRS* deleted mutation on mortality of *S. aureus* bacteremia in a mouse model of blood infection. The cohorts of Balb/C mice were infected with the bacterial cells (approximately 5 × 10^7^ CFU per mouse) from the mid-log phase of culture: wild type *S. aureus* USA300 CA-MRSA 923, *coa* null mutant, *saeRS* null mutant, or *coa*/*seaRS* double null mutant. The mice were monitored daily, and the survival rate of infected mice was calculated every 24 h for 7 days after infection. From 24, 48, and 72 h after infection, the survival rate was the survival rate of Trial I plus Trial II (n = 11) mice infected with each bacterial strain. After 3 days of infection, the survival rate at each time point was calculated using the Trial II mice (n = 5).

**Figure 2 pathogens-10-01396-f002:**
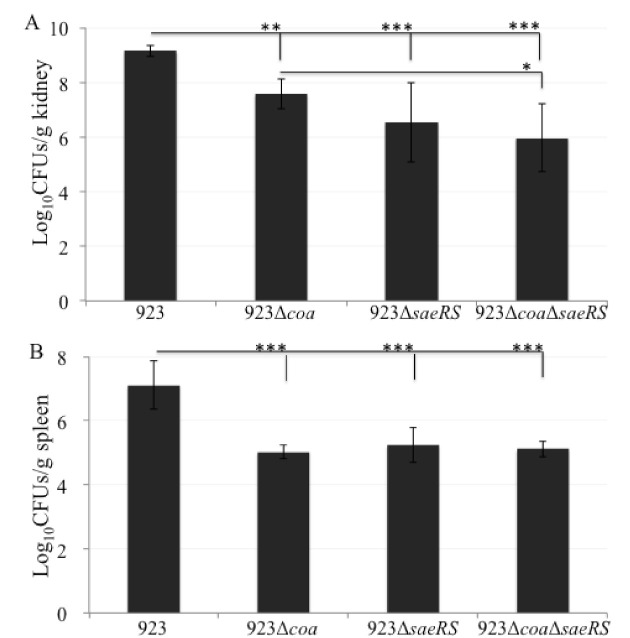
Impact of *coa*, *saeRS*, or *coa*/*saeRS* deleted mutation on the bacterial burden of the kidneys (**A**) and spleens (**B**) isolated from the infected mice in a mouse model of blood infection. The cohorts of mice (n = 6) were sacrificed, and the kidneys and spleens were isolated 3 days after infection. The spleens and partial kidneys were weighed, homogenized, diluted in PBS, and plated onto TSA plates for viable CFU. The difference of log_10_CFUs/gram tissue was statistical analysis with *t*-test. The symbol * represents significance *p* < 0.05. The symbol ** represents *p* < 0.01, and the symbol *** represents *p* < 0.001.

**Figure 3 pathogens-10-01396-f003:**
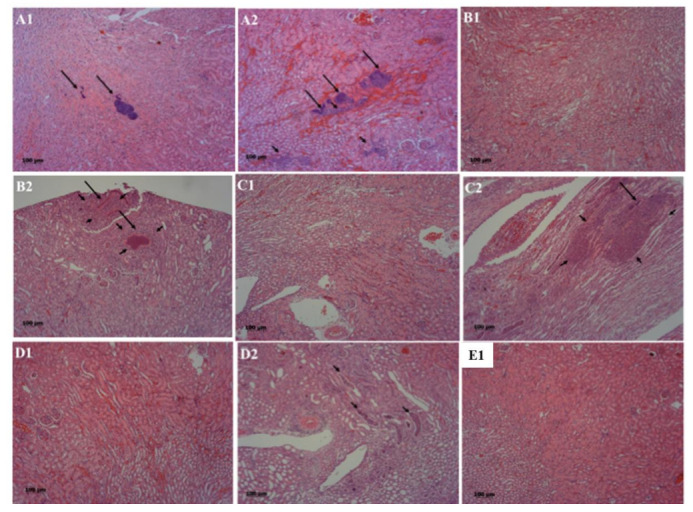
Impact of *coa*, *saeRS*, or *coa*/*saeRS* deleted mutation on the histopathology of the kidneys isolated from the mice infected with USA300 CA-MRSA 923. The kidneys were isolated from the infected mice 3 days (3 dpi with wild-type control and mutant), 4 days (4 dpi with wild-type control), and 7 days (7 dpi with mutant) after infection, and half the kidney was fixed with 10% neutral buffered formalin and embedded in paraffin. The fixed kidneys were sliced and stained with hematoxylin and eosin. In the bacterial cells, pathological damages including infiltration of neutrophils were observed under the microscope. (**A1**) (3 dpi) and (**A2**) (4 dpi) of the kidneys infected with wild type *S. aureus* USA300 CA-MRSA 923; (**B1**) (3 dpi) and (**B2**) (7 dpi) of the kidneys infected with the Coa null mutant; (**C1**) (3 dpi) and (**C2**) (7 dpi) of the kidneys infected with the SaeRS null mutant; (**D1**) (3 dpi) and (**D2**) (7 dpi) of the kidneys infected with the Coa/SeaRS double null mutant; (**E1**) of the kidney uninfected control. Except for slide (**A2**), each slide is representative of two kidneys from each treatment. Short arrow: neutrophil infiltration; long arrow: bacterial communities. dpi: day post infection.

**Figure 4 pathogens-10-01396-f004:**
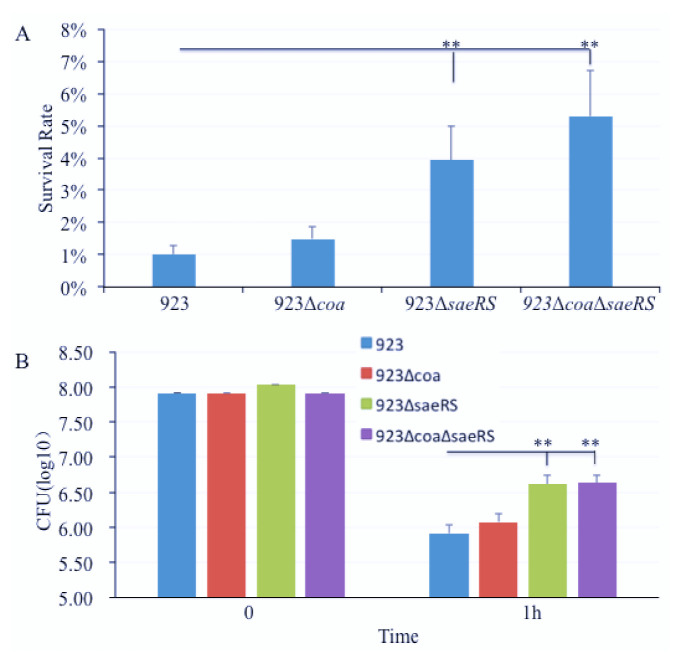
Effect of *coa*, *saeRS*, or *coa*/*saeRS* deleted mutation on survival of *S. aureus* in mouse blood in vitro. (**A**) Percent survival and (**B**) the bacterial number of wild type *S. aureus* USA300 CA-MRSA 923, *coa* null mutant, *saeRS* null mutant, and *coa*/*seaRS* double null mutant in freshly collected heparinized mice blood. The bacterial cells were harvested from the mid-log phase of culture, diluted, inoculated into the blood, and incubated at 37 °C in a rotisserie incubator. Percent survival = (CFU time point/CFU initial input) × 100. The data represent the means ± SEM of at least six independent experiments. The symbol ** represents *p* < 0.01.

**Figure 5 pathogens-10-01396-f005:**
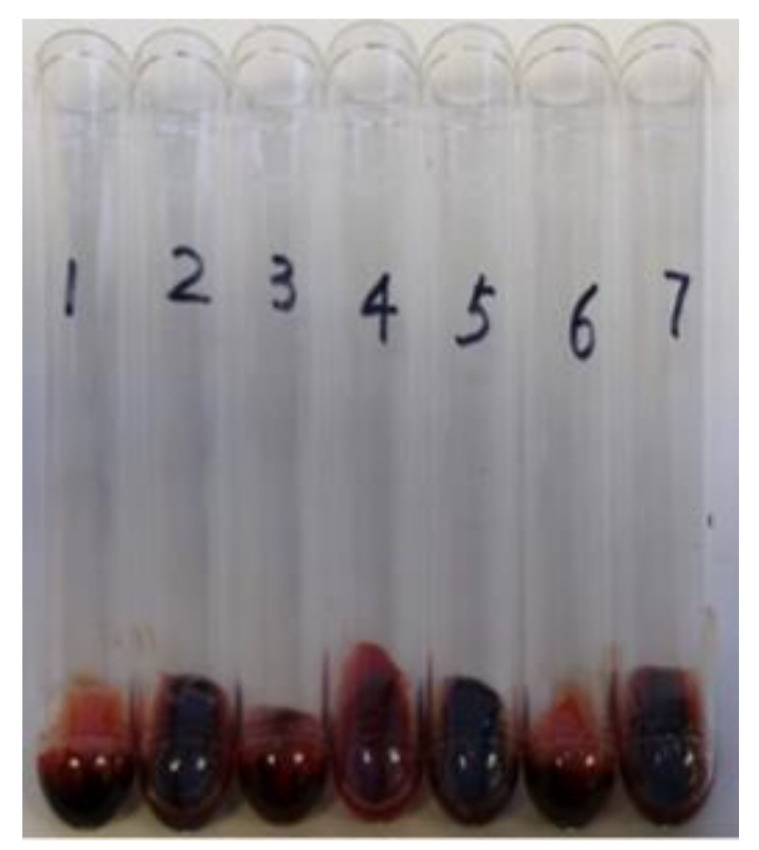
Effect of *coa*, *saeRS*, or *coa*/*saeRS* deleted mutation on blood coagulation of CA-MRSA 923. 1: 923/pYH4, 2: 923Δ*coa*/pYH4, 3: 923Δ*coa*/pYH4-*coa*, 4: 923Δ*saeRS*/pYH4, 5: 923Δ*coa*Δ*saeRS*/pYH4, 6: 923Δ*coa*Δ*saeRS*/pYH4-*coa*, 7: 923Δ*coa*Δ*saeRS*/pYH4-*saeRS*.

**Table 1 pathogens-10-01396-t001:** Mice groups for *S. aureus* infection.

Bacterial Strain	Number of Mice in Trial I	Number of Mice in Trial II
923	6	5
923Δ*coa*	6	5
923Δ*saeRS*	6	5
923Δ*coa*Δ*saeRS*	6	5
Negative control (PBS)	6	5

**Table 2 pathogens-10-01396-t002:** Bacterial strains used in this study.

Strain	Relevant Characteristics	Reference
923	USA300 CA-MRSA	[9]
923Δ*saeRS*	923 *saeRS* deletion mutant	[35]
923Δ*coa*	923 *coa* deletion mutant	[35]
923Δ*coa*Δ*saeRS*	923 *coa* and *saeRS* double deletion mutant	[35]
923/pYH4	923 carrying plasmid pYH4, Erm^R^	[35]
923Δ*coa*/pYH4	923 *coa* deletion mutant with empty plasmid	[35]
923Δ*coa*/pYH4-*coa*	*coa* complementary strain, Erm^R^	[35]
923Δ*saeRS*/pYH4	923 *saeRS* deletion mutant with empty plasmid	[35]
923Δ*saeRS*/pYH4-*coa*	Erm^R^	[35]
923Δ*coa*Δ*saeRS*/pYH4-*coa*	Erm^R^	[35]
923Δ*coa*Δ*saeRS*/pYH4-*saeRS*	Erm^R^	[35]

## Data Availability

The data presented in this study are available in the main text, figures, tables.

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
