# Peer review of "Contribution of Coagulase and Its Regulator SaeRS to Lethality of CA-MRSA 923 Bacteremia"

_pathogens, 2021, doi:10.3390/pathogens10111396_

Round 1
Reviewer 1 Report
The manuscript by Liu et al. found coagulase plays an important role in the lethality of CA-MRSA923 bacteremia in a SaeRS-dependent manner. This topic is very important for the further control and therapy development of Staphylococcus aureus.
I think this manuscript will be stronger if the following points can be explored or clarified:
- Table 2 is not standard three-line grid.
- The X axis can be used x days instead of xx hours.
- The format of References is not consistent.
Author Response
Responses to reviewer 1 comments
We thank the reviewer's efforts for reviewing our manuscript. We revised the manuscript based on the reviewer's comments and suggestions in the revised manuscript.
Point 1. Table 2 is not a standard three-line grid.
Response 1: Table 2 was modified as suggested in the revised manuscript.
Point 2. The X axis can be used x days instead of xx hours.
Responses 2: The X-axis was modified using the day as a unit in the revised manuscript.
Point 3. The format of References is not consistent.
Response 3: The formant of references was modified using a consistent format in the revised manuscript.
Reviewer 2 Report
This article deals with a very important topic , for which reliable data are needed. Interesting data are presented that deserve to be taken into account.
Some questions and observations:
I do not understand the sentence referring to Figure 1. It is confusing. Can you please reformulate.
I suggest and encourage an analysis between the expression of other coagulation factors, such as Von Willebrand factor binding protein. And assess if there is any association.
Material and methods part-4.2 . line 283-285- for how long the incubation was done.
Thank you!
Regards!
Author Response
Responses to Reviewer 2 comments
This article deals with a very important topic, for which reliable data are needed. Interesting data are presented that deserve to be taken into account.
We appreciate the reviewer's time in reviewing our manuscript and comments and suggestions. We made our efforts to revise the manuscript based on the suggestions.
Some questions and observations:
Point 1. I do not understand the sentence referring to Figure 1. It is confusing. Can you please reformate?
Response 1: We reformated the sentence on the top of page 3
Point 2. I suggest and encourage an analysis between the expression of other coagulation factors, such as Von Willebrand factor binding protein. And assess if there is any association.
Response 2: Thank you for your constructive suggestions. In this paper, we aimed to determine whether the coagulase and its direct regulator SaeRS contribute to the lethality of CA-MRSA 923 strain in vivo because our previous studies demonstrated that the deletion of coa or SaeRS increased the bacterial survival in human blood (Guo et al. Front Cell Infect Microbiol.;7:204. doi: 10.3389/fcimb.2017.00204). In addition, the deletion of coa totally eliminated the bacterial ability of coagulation in CA-MRSA 923 strain but not WCUH29 strain (Guo et al. Front Cell Infect Microbiol.;7:204. doi: 10.3389/fcimb.2017.00204), which is the reason why we performed animal studies using CA-MRSA 923 strain in this study. We were focused on coagulase in this study, however, we will examine the expression of coagulation factors, including the Von Willebrand factor between different strains as suggested in future studies.
Point 3. Material and methods part-4.2 . line 283-285- for how long the incubation was done.
Response 3: we added more detailed information on the incubation time in the section of 4.1 and 4.2 on page 9 in the revised manuscript.
Reviewer 3 Report
Staphylococcus aureus is an important human and animal pathogen and causes numerous infections with high mortality. It produces a plethora of virulence factors, and several regulatory circuits are known to control their synthesis. The paper of Dr. Ying Liu et al describes influence of coagulase on lethality, histological pictures and blood survival of S. aureus using strains with deleted coagulase-coding gene coa and its transcriptional regulator saeRS. The manuscript appears to represent a daughter-paper of a previous study from the same scientific group (doi: 10.3389/fcimb.2017.00204).
Major points.
- The obtained data are of modest novelty, bearing in mind quite impressive number of papers on the topic in the last two decades. The results are low in numbers and look confusing.
- Methods may be simple and needs to be improved.
- No studies were performed on characterization of the made strains in respect to transcription/translation of the deleted genes/mRNA (coa and saeRS) as well as of some other genes regulated by saeRS. Polar effects on transcription of unstudied here genes are highly possible. These important investigations have not been performed even in the previous paper from the same group (doi: 10.3389/fcimb.2017.00204), in which the engineering of the mutated strains was described. Therefore, I conclude that the mutant strains have not been sufficiently characterized and the conclusions should be accepted with caution. This holds true especially for finding that deletion of a regulatory gene decreases infection (quite reasonable) but increases blood survival (not very reasonable). Really solid and diversified experiments are needed here to exclude some sort of scientific mistakes.
- In contrast to the previous paper from the same group (see above) no complementation studies (i.e., deleted strains back transformed with the gene that has been deleted) have been performed.
- It looks puzzling that deletion of a single coa produces effect on mice lethality comparable to deletion of saeRS, bearing in mind that the latter protein controls transcription of a broad panel of virulence factors including highly toxic ones (e.g. HlyA).
- Chapter 4.2. This is not “The phagocytosis assays…” Phagocytosis assay is performed with isolated phagocytes; here is just incubation of bacteria in blood. The latter phenomenon is multifactorial, and it is not clear what has been actually studied.
Minor points.
- Line numbers would be desirable.
- Table 2 is actually Table 1 as it appears in the manuscript. It is non-informative in general and can be substituted by a single sentence with quantity of mice used in the experiments.
- Figure 1. What is the reason for very high standard deviations for mutants in panel A?
- Chapter 2.1. “The deletion mutation of coa significantly alleviated the lethality of CA-MRSA 923 in a murine blood infection” and Chapter 2.2. “The saeRS single knockout or coa/saeRS double knockout significantly eliminated the lethality of CA-MRSA 923 in a murine blood infection”. As seen, all tested mutations “significantly eliminated/alleviated the lethality…” Therefore, no need to split them into separate chapters.
- Figure 3. “dpi” in histology sections probably means days post infection and not dots per inch?
- Figure 4. Data on panel A and B seem to be in principle identical. One graph would be enough.
Author Response
Responses to reviewer 3 comments
Thank you so much for spending the time to review our manuscript and give constructive comments and suggestions. We revised the manuscript as you suggested.
Point 1. The obtained data are of modest novelty, bearing in mind quite impressive number of papers on the topic in the last two decades. The results are low in numbers and look confusing.
Response 1: We agree with the reviewer's comments. However, we are the first to examine the role of coagulase and its regulator SaeRS in the lethality of the USA300 CA-MRSA 923 strain in bacteremia.
Point 2. Methods may be simple and needs to be improved.
Response 2: We modified the Materials and Methods section of 4.1 and 4.2 on page 9 in the revised manuscript.
Point 3. No studies were performed on characterization of the made strains in respect to transcription/translation of the deleted genes/mRNA (coa and saeRS) as well as of some other genes regulated by saeRS. Polar effects on transcription of unstudied here genes are highly possible. These important investigations have not been performed even in the previous paper from the same group (doi: 10.3389/fcimb.2017.00204), in which the engineering of the mutated strains was described. Therefore, I conclude that the mutant strains have not been sufficiently characterized and the conclusions should be accepted with caution. This holds true especially for finding that deletion of a regulatory gene decreases infection (quite reasonable) but increases blood survival (not very reasonable). Really solid and diversified experiments are needed here to exclude some sort of scientific mistakes.
Response 3: We didn’t examine the impact of the deletion of saeRS on gene expression because it has been reported that in these same strains, SaeRS regulates multiple virulence factors and pathogenicity in different mouse infection models (see Montgomery et al, 2010 PloS One). Moreover, we have performed diagnostic PCR, sequenced the flanking regions of the coa and seaRS knockout mutants, and complementation studies in our previous studies (Guo et al Front Cell Infect Microbiol.;7:204. doi: 10.3389/fcimb.2017.00204.). In addition, we have reported the impact of the null mutation of saeRS on gene expression, including coa, hla, etc in WCUH29 strain (Liang et al Infect Immun. 2006, 74:4655-4665). Surprisingly, we found that the null mutation of SaeRS increased the ability of WCUH29 to survive in human blood but the introduction of pYH4-coa in the saeRS deletion mutant could eliminate this phenomenon. We used the same approach to create coa, saeRS, and coa/saeRS deletion mutants and demonstrated that the null mutation of Coa, SaeRS, or Coa/SaeRS eliminated the coagulation function in the 923 strain as well as performed a series of complementation studies, see detailed information in Table 3 in our previous publication (Guo et al Front Cell Infect Microbiol.;7:204. doi: 10.3389/fcimb.2017.00204). We agree that without whole-genome sequencing analysis we cannot exclude that off-target mutations may occur during the gene knockout procedures. Due to the difference in expression levels of coagulation factors, including Coa and vWbp among different MRSA strains, we cannot conclude that the importance of coagulase for CA-MRSA 923 strain-caused bacteremia is universally applicable to other CA-MRSA strains.
Point 4. In contrast to the previous paper from the same group (see above) no complementation studies (i.e., deleted strains back transformed with the gene that has been deleted) have been performed.
Response 4: We agree with the reviewer's comment. We didn't perform complementation studies in vivo because feeding animals with a broad-spectrum antibiotic, erythromycin for at least 7 days to maintain the plasmid DNA of bacteria would complicate the experimental results. However, we have performed complementation studies in vitro, including coagulation and bacterial survival studies in human blood, which were published in our previous paper ( Guo et al Front Cell Infect Microbiol.;7:204. doi: 10.3389/fcimb.2017.00204). We performed complementation studies in blood coagulation assays, and our results further indicated the importance of coa and its regulator SaeRS for blood coagulation of CA-MRSA 923 in the revised manuscript (see Figure 5).
Point 5. It looks puzzling that deletion of a single coa produces effect on mice lethality comparable to deletion of saeRS, bearing in mind that the latter protein controls transcription of a broad panel of virulence factors including highly toxic ones (e.g. HlyA).
Response 5: We totally agree with the reviewer's comments. Our results in Figure 1 and Figure 3 also indicate that the deletion of saeRS attenuated much more virulence of the MRSA 923 strain compared to the deletion of coa, although the difference isn't significant. We also modified the abstract and discussion to emphasize the key role of SaeRS in the lethality of CA-MRSA 923 bacteremia and Coagulase is one of the virulence factors that is regulated by SaeRS and contributes to the pathogenicity of CA-MRSA 923.
Point 6. Chapter 4.2. This is not “The phagocytosis assays…” Phagocytosis assay is performed with isolated phagocytes; here is just incubation of bacteria in blood. The latter phenomenon is multifactorial, and it is not clear what has been actually studied.
Response 6: We changed the phagocytosis assays to "survival assays in blood" in the revised manuscript.
Minor points.
Point 7. Line numbers would be desirable.
Response 7: We agree with the reviewer's suggestion but couldn't add the Line numbers in the current word format of the revised manuscript.
Point 8. Table 2 is actually Table 1 as it appears in the manuscript. It is non-informative in general and can be substituted by a single sentence with quantity of mice used in the experiments.
Response 8: We switched the table number in the revised manuscript.
Point 9. Figure 1 (2?). What is the reason for very high standard deviations for mutants in panel A?
Response 9: It is possibly due to the weight variation of isolated kidneys from mice.
Point 10. Chapter 2.1. “The deletion mutation of coa significantly alleviated the lethality of CA-MRSA 923 in a murine blood infection” and Chapter 2.2. “The saeRS single knockout or coa/saeRS double knockout significantly eliminated the lethality of CA-MRSA 923 in a murine blood infection”. As seen, all tested mutations “significantly eliminated/alleviated the lethality…” Therefore, no need to split them into separate chapters.
Response 10: We understood the reviewers' comments. In order to clearly present the results to the readers, we separated coa and saeRS into two chapters
Point 11. Figure 3. “dpi” in histology sections probably means days post infection and not dots per inch?
Response 11: We added that "dpi" means "day post-infection" in Figure 3 legend in the revised manuscript.
Point 12. Figure 4. Data on panel A and B seem to be in principle identical. One graph would be enough.
Response 12: We keep both panels because panel A represents the percentage of bacterial survival and panel B represents the actual CFUs of surviving bacteria in blood.
Reviewer 4 Report
In this article the authors look at the effect of deletion of either coagulase (coa gene) or a regulator saeRS using an in vivo blood infection model in mice. Similar work has been done using the same strains but in different infection models that came to similar conclusions- that both genes are important for virulence (eg Montgomery et al., 2010 for the role of SaeRS).
Overall, I think the paper could be improved by making the conclusions a bit more rounded: SaeRS has an effect on several virulence factors, not just coagulase, and so it is not clear that the results in vivo should be attributed solely to reducing coagulase. Further, it was not clearly demonstrated that coagulase expression is completely absent in the saeRS mutant, or what its levels were?
Specific comments:
- Deletion of saeRS results in the reduced expression of a number of virulence factors in USA300 (see Montgomery et al, 2010 PloS One for example) and in other strains: Rogasch K, Ruhmling V, Pane-Farre J, Hoper D, Weinberg C, et al. (2006) Influence of the two-component system SaeRS on global gene expression in two different Staphylococcus aureus J Bacteriol 188: 7742–7758.
- “These results clearly indicate that SaeRS is a critical virulence regulator and the lethality of CA-MRSA923-caused bacteremia is mostly attributable to the SaeRS regulation of coagulase in the mouse model of blood infection.” It is not obvious to me that the only reason is the coagulase? I would rephrase it as a major contributor. Also, it is not directly shown that in the mouse blood infection model that coa expression is reduced in the saeRS mutant (although it seems highly likely). Is this data available? Are there references that could back up this assertion?
- Figure 3: the legend is missing an explanation of what the arrows point to, and why some are different sizes. This is present in the text, but should really be in the legend as well.
- Discussion: it has also has been shown in these same strains that SaeRS affects virulence, but in different mouse infection models. Perhaps this needs more emphasis.
- Line 253: conversely rather than controversy. The authors might want to look over their use of controversy throughout? There were a few other typos as well, but on the whole well written.
- Methods: in the Strains list, there are the strains carrying plasmids for complementing the mutant phenotypes, yet these aren’t used in any of the experiemnts that I could see, and are only mentioned in the discussion paragraph about drawbacks of the experimental methods.
- The paragraph on the limitations of the project could be improved- it does not read particularly well.
Author Response
Response to reviewer 4 comments
In this article the authors look at the effect of deletion of either coagulase (coa gene) or a regulator saeRS using an in vivo blood infection model in mice. Similar work has been done using the same strains but in different infection models that came to similar conclusions- that both genes are important for virulence (eg Montgomery et al., 2010 for the role of SaeRS).
Overall, I think the paper could be improved by making the conclusions a bit more rounded: SaeRS has an effect on several virulence factors, not just coagulase, and so it is not clear that the results in vivo should be attributed solely to reducing coagulase. Further, it was not clearly demonstrated that coagulase expression is completely absent in the saeRS mutant, or what its levels were?
Thank you so much for spending the time to review our manuscript and giving constructive comments and suggestions. We revised the manuscript as you suggested.
Specific comments:
Point 1. Deletion of saeRS results in the reduced expression of a number of virulence factors in USA300 (see Montgomery et al, 2010 PloS One for example) and in other strains: Rogasch K, Ruhmling V, Pane-Farre J, Hoper D, Weinberg C, et al. (2006) Influence of the two-component system SaeRS on global gene expression in two different Staphylococcus aureus J Bacteriol 188: 7742–7758.
Response 1: We agree with the reviewer's comments. We included reference 29 on page 8, which is highlighted in yellow color in the revised manuscript.
Point 2. “These results clearly indicate that SaeRS is a critical virulence regulator and the lethality of CA-MRSA923-caused bacteremia is mostly attributable to the SaeRS regulation of coagulase in the mouse model of blood infection.” It is not obvious to me that the only reason is the coagulase? I would rephrase it as a major contributor. Also, it is not directly shown that in the mouse blood infection model that coa expression is reduced in the saeRS mutant (although it seems highly likely). Is this data available? Are there references that could back up this assertion?
Response 2: We agree with the reviewer's comments. We changed the title of the manuscript, modified the abstract, results on page 5, and discussion on page 7&9, which are highlighted in yellow color, to emphasize the key role of SaeRS in the lethality of CA-MRSA 923 in bacteremia in a mouse model of blood infection and coagulase is a major contributor. We don't have data to show that the expression of Coa is down-regulated in the mouse blood infected with SaeRS knockout mutant.
Point 3. Figure 3: the legend is missing an explanation of what the arrows point to, and why some are different sizes. This is present in the text, but should really be in the legend as well.
Response 3: We added the explanations in the legend of Figure 3 in page 6 in the revised manuscript.
Point 4. Discussion: it has also has been shown in these same strains that SaeRS affects virulence, but in different mouse infection models. Perhaps this needs more emphasis.
Response 4: We agree with the reviewer's comments and added a sentence on page 8 and included reference 9 in the revised manuscript.
Point 5. Line 253: conversely rather than controversy. The authors might want to look over their use of controversy throughout? There were a few other typos as well, but on the whole well written.
Response 5: We changed "controversy" to "conversely" and corrected typos throughout the revised manuscript.
Point 6. Methods: in the Strains list, there are the strains carrying plasmids for complementing the mutant phenotypes, yet these aren’t used in any of the experiemnts that I could see, and are only mentioned in the discussion paragraph about drawbacks of the experimental methods.
Response 6: We did some complementation studies in blood coagulation using the strains carrying the plasmid in the revised manuscript.
Point 7. The paragraph on the limitations of the project could be improved- it does not read particularly well.
Response 7: We modified the paragraph on the limitations of current studies as suggested on page 9 in the revised manuscript.
Round 2
Reviewer 3 Report
I would like to thank the authors of the current paper for answering my questions and responding to my comments. However, I am not satisfied with both. As I stated in my previous Reviewer’s letter, major problems with the manuscript are that it contains insufficient volume of data to draw definite conclusion and these data have been obtained with the insufficiently characterized strains using old-fashioned methods. In their responses, the authors say that they “agree” or even “totally agree” with my critics, but in essence, little has been changed in the revised copy. Preparing my previous Reviewer’s letter, I understood that my comments could not be responded adequately within the framework of the current manuscript. Many additional experiments are needed.
Major point 1. I have said that “The results are low in numbers and look confusing”. And response is “However, we are the first to examine the role…”. The point was not whether you are the first or the second. The point was that the data are not solid enough.
Major point 2. I have said that a poor arsenal of very simple methods has been used. And the response was “We modified the Materials and Methods section of 4.1 and 4.2 on page 9…”. I did not ask for re-phrasing the methods’ descriptions, I did ask for better methodology in the study.
Major point 3. I have said that the study has been performed with under-characterized strains. The authors disagree and tell that the strains had been already characterized in the paper of Montgomery et al and Liang et al. The problem is that in the cited references, the OTHER strains were implemented, and I would like to see the characteristics of these particular, used in the current study strains.
Major point 4. I have specified that complementing in vivo experiments would be desirable to confirm specificity of the lethal effect. The authors answered that “…feeding animals with a broad-spectrum antibiotic, erythromycin for at least 7 days to maintain the plasmid DNA of bacteria would complicate the experimental results”. I partially agree, but still, it worth trying to perform such an experiment without antibiotic, because coa+, or coa/saeRS+ variants seem to be more virulent and selective pressure might help keeping the plasmids inside bacteria in vivo.
Major point 5. I have said that it looks strange that “the deletion of a single coa produces effect on mice lethality comparable to deletion of saeRS…” The response is yes, we can see it on figures. For me, the problem is how can one explain it? What kind of experiments should be done to explain it? Why have these experiments not been performed? Actually, this is the core of my concern.
Major point 6. I have said that incubation of bacteria in blood is not a phagocytosis assay. The authors seem to agree with this, but in the revised version one can read “phagocytosis assay” on pages 202, 299 and 301.